# Degradation and Lifetime Prediction of Epoxy Composite Insulation Materials under High Relative Humidity

**DOI:** 10.3390/polym15122666

**Published:** 2023-06-13

**Authors:** Jielin Ma, Yan Yang, Qi Wang, Yuheng Deng, Malvern Yap, Wen Kwang Chern, Joo Tien Oh, Zhong Chen

**Affiliations:** 1SP Group–NTU Joint Laboratory, School of Electrical and Electronic Engineering, Nanyang Technological University, 50 Nanyang Avenue, Singapore 639798, Singapore; 2School of Materials Science and Engineering, Nanyang Technological University, 50 Nanyang Avenue, Singapore 639798, Singapore; 3Singapore Power Group, Singapore 349277, Singapore

**Keywords:** epoxy composite, material properties, lifetime prediction model, degradation mechanism

## Abstract

Insulation failure of composite epoxy insulation materials in distribution switchgear under the stress of heat and humidity is one of the leading causes of damage to switchgear components. This work prepared composite epoxy insulation materials by casting and curing a diglycidyl ether of bisphenol A (DGEBA)/anhydride/wollastonite composite system, and performed material accelerated aging experiments under three conditions: 75 °C and 95% relative humidity (RH), 85 °C and 95% RH, and 95 °C and 95% RH. Material, mechanical, thermal, chemical, and microstructural properties were investigated. Based on the IEC 60216-2 standard and our data, tensile strength and ester carbonyl bond (C=O) absorption in infrared spectra were chosen as failure criteria. At the failure points, the ester C=O absorption decreased to ~28% and the tensile strength decreased to 50%. Accordingly, a lifetime prediction model was established to estimate material lifetime at 25 °C and 95% RH to be 33.16 years. The material degradation mechanism was attributed to the hydrolysis of epoxy resin ester bonds into organic acids and alcohols under heat and humidity stresses. Organic acids reacted with calcium ions (Ca^2+^) of fillers to form carboxylate, which destroyed the resin-filler interface, resulting in a hydrophilic surface and a decrease in mechanical strength.

## 1. Introduction

Electrical insulation materials protect switchgear components from transient voltages caused by sudden energy surges such as lightning, arcing, heavy inductive loads, and electrostatic discharges. Significantly, epoxy composite insulation materials are widely applied to switchgear insulation due to their excellent electrical breakdown strength, mechanical toughness, good adhesion to metal conductors and dielectric materials, heat resistance, versatility, low cost, and ease of manufacture [1,2,3]. However, epoxy composite materials in switchgear can be severely degraded under long-term exposure to high humidity, high temperature, and strong electric fields, leading to insulation failure [4,5,6]. It is critical to study the degradation of epoxy composite insulation materials and establish a reliable lifetime prediction model for key components of switchgear.

The aging performance of silicone rubber-based insulation materials in various application environments (high humidity, elevated temperature, low temperature icing, acidic, salt fog, corona discharge, strong radiation environments) has been well studied, including their electrical, physical, and chemical properties [7,8,9,10,11,12,13,14,15,16,17,18]. The deterioration behaviour of epoxy resin-based insulation materials under thermal, thermos-oxidative, hygrothermal, electrical-thermal, and alternating current (AC) field aging was researched with regard to mass loss, relative permittivity, electrical breakdown voltage, dielectric loss, flashover voltage, surficial leakage current, contact angle of hydrophobicity, glass transition temperature, the infrared spectrum, the X-ray energy spectrum, dynamic mechanical analysis, and tensile properties [5,19,20,21,22,23].

In brief, the aging performance change in polymer-based insulation materials can be categorised into four main categories: physical, chemical, thermal, and electrical [7,8,20,22,23,24]. The physical performance includes deformation, fracture, roughness, colour, hydrophobicity, hardness, and tensile strength. The chemical performance includes the molecular functional groups’ composition and the elemental composition. The thermal performance includes the glass transition temperature, thermal expansion, thermal conductivity, and thermal decomposition. The electrical performance includes alterations in material dielectric loss, electric conductivity, and breakdown voltage. These performances are affected by factors which include, but are not limited to, temperature, humidity, voltage, ultraviolet irradiation, acidity, alkalinity, ozone, and nitrogen oxides. Table 1 summarizes the various material properties, characterization techniques, and relevant industrial standards for switchgear insulation material degradation studies.

According to the international standards of the IEC (International Electrotechnical Commission) 60505 [45], the hygrothermal aging mechanism of epoxy composite insulation materials includes moisture absorption, hydrolysis of epoxy resin ester bonds, polymer molecular chain breakage under thermal activation, cracking of epoxy composite insulation materials, and loss of adhesion between the epoxy resin and inorganic fillers due to thermal expansion. Mechanistic analyses of these key factors affecting degradation rates, together with lifetime prediction models, are essential for the reliable operation of electrical power systems. Recently, some interesting discoveries have been made regarding the aging process and degradation mechanisms of epoxy resin composites. Ghabezi et al. [46] evaluated the aging process of glass fabric-reinforced epoxy resin and carbon fabric-reinforced epoxy resin aged in hot artificial salt water. Data indicated that glass fabric-reinforced epoxy resin has higher water absorption than carbon fabric-reinforced epoxy resin. The silica and metal oxides in the glass fabric are likely responsible for the higher water uptake. Liu et al. [47] investigated the hygrothermal aging behaviour of carbon fiber-reinforced epoxy composites. They found that the water absorption of the epoxy composite fits into Fick’s model. The absorbed water molecules reacted with the resin matrix, plasticized the resin matrix, and damaged the interfaces between the resin matrix and carbon fibers, resulting in a significant decrease in the short-beam shear strength and glass transition temperature.

Epoxy composite insulation is critical for safe and reliable power supply. The cost of an insulation failure can be enormous, and the damage can be extensive. To reduce the risk, it is necessary to systematically study the degradation behaviour of epoxy composite insulation materials to replace unhealthy components before failure occurs. According to the literature review, there is a lack of systematic degradation studies of aged epoxy composite insulation materials in terms of their mechanical, thermal, and chemical properties; additionally, research on the effects of hygrothermal aging on material properties for the purpose of improved service life prediction is also lacking. The novelty of this research is that standard testing techniques have been implemented to quantitatively assess the degradation behaviour of epoxy composite insulation materials through accelerated aging experiments and various material characterization techniques. In this work, the degradation and lifetime prediction of epoxy composite insulation materials under high humidity were studied. First, epoxy composite insulation materials in four shapes (dumbbell-shaped, rectangular, small circular, and large circular) were prepared by casting and curing a two-part epoxy composite system in designed silicone rubber moulds. Next, material accelerated aging experiments were performed under three conditions: 75 °C and 95% RH, 85 °C and 95% RH, and 95 °C and 95% RH. Material properties at different aging times were characterized by tensile testing, laser flash, TMA, DMA, weighing balance, dimensional measurements, FTIR, TGA, contact angle testing, and FESEM-EDS. Data analysis of tensile strength, infrared spectra, cross-sectional morphology, and contact angles allowed us to quantify the extent of material degradation, obtain lifetime prediction models, and understand degradation mechanisms.

## 2. Materials and Methods

### 2.1. Materials

Huntsman Araldite^®^ casting resin system #229 was purchased from Vitrochem Technology Pte Ltd., Singapore, Singapore. It consists of part A [Araldite^®^ CW 229 CI resin: diglycidyl ether of bisphenol A (DGEBA) 30–50 wt.%, wollastonite CaSiO_3_ ~55 wt.%]; and part B (Aradur^®^ HW 229 CI hardener: anhydride 20–30 wt.%, modified anhydride 10–20 wt.%, filler ~62 wt.%). Henkel Loctite Frekote B-15 mould sealer and Henkel Loctite Frekote 770-NC release agent were purchased from Auzana Industries (S) Pte Ltd. (Singapore, Singapore). Two-component platinum-cured silicone #1530 (part A—platinum hardener; part B—silicone base) was purchased from Shenzhen Rongxingda Polymer Material Co. Ltd., Shenzhen, China. A piece of circular disk (diameter 100 mm, thickness 1.0 mm), three pieces of ASTM D638-14 standard type V dumbbell-shape (thickness 2.5 mm) [25], and three pieces of rectangular master (length 40 mm, width 10 mm, thickness 2.5 mm) were fabricated by Beijing Wauteco Testing Equipment Ltd., Beijing, China. These masters are made of a zinc-aluminum alloy and were used to prepare the silicone moulds for the casting of epoxy composite samples. Glass fiber-reinforced Teflon release films (thickness 80 μm) were purchased from Lazada Singapore (Singapore, Singapore). Figure 1 is a schematic of the standard test samples including their dimensions.

### 2.2. Synthesis of Epoxy Composites

The sample preparation includes preparing silicone rubber moulds and casting epoxy composites in them. The preparation of silicone rubber moulds consists of the following steps: (1) Mix part A and part B of platinum cured silicone #1530 in a weight ratio of 1:1; (2) Degas the mixture in a vacuum chamber; (3) Pour the degassed mixture into a petri dish containing fixed alloy masters; (4) Degas the filled petri dish in a vacuum chamber; (5) Cure the degassed petri dish in an oven (UNE 400, Memmert GmbH + Co.KG, Schwabach, Germany) at 60 °C for 2 h; (6) Take out the silicone rubber moulds from the petri dish; (7) Heat the moulds at 200 °C for an hour to remove moisture and strengthen them; (8) Store the moulds at room temperature with 50% relative humidity (RH) in airtight dry containers.

Following existing research and standards [48,49,50,51,52], the epoxy composites were prepared as follows: (1) Apply the sealer and release agent to the silicone rubber moulds in a well-ventilated fume hood and cure them under appropriate conditions according to the supplier’s technical data sheet; (2) Mix part A and part B of the Huntsman Araldite^®^ casting resin system #229 (weight ratio 1:1) in a clear complete fast-freeze flask (750 mL, Labconco, Kansas City, MO, USA) attached with an adapter to a valve port on a working vacuum oil pump and keep stirring with a magnetic stirrer (MR Hei-Tec, Heidolph Instruments GmbH & Co.KG, Schwabach, Germany). Meanwhile, heat the flask in an oil bath from 25 °C to 65 °C and keep heating at 65 °C for 30 min to remove all air bubbles; (3) Quickly and carefully pour the degassed mixture into silicone moulds that have been preheated in an oven at 100 °C for 1 h. Cover the moulds with glass fiber-reinforced Teflon release film, weighing paper, and a metal block in sequence; (4) Cure the covered mixture in an oven at 100 °C for 4 h, followed by 12 h at 135 °C; (5) Demould samples after the oven cools to room temperature. Condition the samples in a dry container at 25 °C and 50% RH for 16 h. Figure 2 illustrates the epoxy resin networks synthesized from DGEBA epoxy resin and anhydride hardener [53] and their ester bond degradation reaction under hygrothermal aging conditions [54].

### 2.3. Accelerated Aging Experiment

According to a dataset recorded by Changi climate station on Data.gov.sg, the daily mean RH over the last 10 years in Singapore was between 76% and 84.6%, the daily minimum RH was between 57.2% and 67.4%, and the maximum RH was between 90% and 97%. In this study, we chose the most critical humidity as the accelerated aging condition. According to the technical data sheet from the supplier, the glass transition temperature of epoxy composites was between 110 °C and 120 °C. For electrical insulation materials, the maximum test temperature should be below their transition temperatures, including for glass transition, melting, boiling, and crystallization [45]. For structural, core, and adhesive materials, the maximum test temperature must be less than Tg minus 15 K according to IEC 62039 [55]. Based on IEC 60505, a minimum of three temperatures shall be selected to process accelerated aging; thus, we chose 75 °C, 85 °C, and 95 °C as the accelerated aging temperatures. Environmental test chambers (CTC256, Memmert GmbH + Co.KG, Schwabach, Germany) were utilized for accelerated degradation tests, as these chambers are capable of active humidification and dehumidification from 10% RH to 98% RH, along with precise and rapid temperature changes from −42 °C to 190 °C. IEC 60749-5 provides a standard method to process steady-state temperature humidity bias life tests for evaluating the reliability of non-hermetic packaged solid-state devices in humid environments [56]. Referring to standard methods, the accelerated aging tests of epoxy composites were performed under different aging conditions, as shown in Table 2. Figure 3 shows epoxy composite sample preparation for the accelerated aging experiment.

### 2.4. Material Characterizations

#### 2.4.1. Tensile Test

The tensile strength of dumbbell-shape samples was investigated using a universal mechanical testing machine (MTS Criterion Model 43, Eden Prairie, MN, USA) according to ASTM D638-14. Dumbbell-shape specimens were tested at room temperature using a 10 KN load cell and Advantage Wedge Action Grips (MTS, Model #: 10) with a crosshead speed of 1 mm/min. There were five specimens per aging condition timepoint. The average value and standard deviation were calculated.

#### 2.4.2. FTIR

The attenuated total reflection Fourier-transform infrared (ATR-FTIR) absorbance of rectangular samples was scanned using a Fourier-transform infrared spectrometer (Frontier, PerkinElmer, Waltham, MA, USA) and a universal ATR attachment in the wavenumber range of 4000–600 cm^−1^ at 4 cm^−1^ resolution. Both the height and area of infrared absorbance were utilized for quantitative analysis [57,58], while the integrated peak area was preferred over the peak height to reduce the influence of peak broadening [59]. Relative ester C=O absorption in samples was quantified by calculating the ratio of the areas of the two absorption bands (1775–1683 cm^−1^ for the ester C=O bonds and 1522–1485 cm^−1^ for aromatic ring C=C as an internal standard). The ester C=O absorption in unaged samples was defined as 100%. Five specimens were tested per aging condition timepoint. The average value and standard deviation were calculated.

#### 2.4.3. DMA

Glass transition temperature (*T_g_*) was measured using a dynamic mechanical analyzer (DMA Q800, TA Instruments, New Castle, DE, USA). Measurements were performed at a rate of 3 °C/min and a frequency of 1 Hz in the range of 30 °C to 150 °C. Glass transition temperatures of the samples were determined from the maximum value of the tan δ peak. There were five rectangular specimens for each aging condition timepoint. The average value and standard deviation were calculated.

#### 2.4.4. TMA

The coefficient of linear thermal expansion of the epoxy composite was determined using a thermomechanical analyzer (TMA Q400, TA Instruments, New Castle, DE, USA). Measurements were performed from 30 °C to 150 °C at a rate of 5 °C/min using nitrogen as a purge gas at a flow rate of 50 mL/min and a load force of 0.05 N. Three specimens were tested for each aging condition timepoint. Means and standard deviation were calculated.

#### 2.4.5. Laser Flash

Circular samples with a diameter of 12.7 mm and a thickness of approximately 1.5 mm were prepared by casting Huntsman epoxy composite system #229 using silicone rubber moulds. The Vespel reference sample with data (12.7 mm diameter, 1.27 mm thick) was purchased from TA Instruments. Density files of all samples and heat capacity reference files for the Vespel reference sample were created prior to testing. All sample surfaces were coated with a thin layer of graphite to increase laser emissivity and eliminate sample transparency. Thermal conductivity and heat capacity data were determined using a laser flash system (DLF 1200, TA Instruments, New Castle, DE, USA) using argon as a purge gas. The volumes of the samples were determined by the diameter and thickness of the samples. The densities of the samples were then calculated from their mass and volume. Three specimens were tested for each aging condition timepoint. Means and standard deviations were reported.

#### 2.4.6. TGA

Material decomposition temperatures and weight loss were measured using thermogravimetric analyzers (TGA Q500, TA Instruments, New Castle, DE, USA). Measurements were performed from 30 °C to 900 °C at a rate of 10 °C/min using nitrogen as a purge gas with a sample purge flow rate of 60 mL/min and an equilibrium purge flow rate of 40 mL/min.

#### 2.4.7. Contact Angle

Static optical contact angles were measured using a contact angle goniometer (OCA 15Pro, Dataphysics Instruments, Filderstadt, Germany) to evaluate the contact angle between the edge of a single deionized water droplet and the surface of a solid material.

#### 2.4.8. FESEM-EDS

The microstructure images and elemental composition of the materials were investigated using a field emission scanning electron microscope (FESEM JSM-7600F, JEOL, Tokyo, Japan) equipped with an energy dispersive spectrometer (EDS). The filler was purified from the uncured Huntsman epoxy composite system #229 by a series of washes and precipitations using three kinds of solvents, including 95% ethanol, 99% acetone, and distilled water, and dried in a 50 °C oven overnight before imaging. The sample surface was polished using polishing machines (Forcipol 1V, Metkon, Bursa, Turkey) for element composition analysis. The sample surface was coated with a thin platinum layer in argon using an auto fine coater (JFC-1600, JEOL, Tokyo, Japan).

#### 2.4.9. Mass Change

The mass change in big disk samples (diameter 100 mm, thickness 1.5–2.5 mm) was determined by weighing their initial mass and the mass after the accelerated aging experiment, and this change was expressed as a percentage of the initial mass.

## 3. Results and Discussion

### 3.1. Summary of Material Properties Change with Aging

The material properties over time under different aging conditions were investigated by experimental techniques, as shown in the Appendix A. FTIR, tensile testing, and DMA of unaged samples indicated the successful synthesis of epoxy composites with a tensile strength of 69.1 ± 6.4 MPa and a glass transition temperature of 116 ± 6 °C (Appendix A). For the tensile strength of epoxy composites, the reduction was 50.5% ± 9.1% after 1000 h of aging at 95 °C and 95% RH, 50.1% ± 7.2% after 2012 h of aging at 85 °C and 95% RH, and 28.9% ± 10.9% after 2013 h of aging at 75 °C and 95% RH. Obviously, the material mechanical strength decreased with the extension of the aging time, and the degradation speed accelerated with the increase in the aging temperature. For ester C=O absorption, the reduction was 77.4% ± 0.6% after 1000 h of aging at 95 °C and 95% RH, 76.4% ± 7.2% after 2012 h of aging at 85 °C and 95% RH, and 42.6% ± 12.9% after 2013 h of aging at 75 °C and 95% RH. The decrease in the C=O absorption of epoxy composites indicated that the epoxy ester bonds were broken during aging, disrupting the epoxy resin network, and compromising mechanical strength. For the glass transition temperature, the reduction was 12.1% ± 0.9% after 1000 h of aging at 95 °C and 95% RH, 9.5% ± 3.4% after 2012 h of aging at 85 °C and 95% RH, and 13.8% ± 8.6% after 2013 h of aging at 75 °C and 95% RH. For mass change, the data showed an increase of 0.77% ± 0.10% after 1000 h of aging at 95 °C and 95% RH, an increase of 0.86% ± 0.04% after 2012 h of aging at 85 °C and 95% RH, and an increase of 0.78% ± 0.20% after 2013 h of aging at 75 °C and 95% RH. The mass of epoxy composites increased due to water absorption during accelerated degradation tests at 95% RH.

Most thermal properties (thermal conductivity, heat capacity, coefficient of linear thermal expansion, mass loss at 900 °C under nitrogen, and decomposition temperature) did not change significantly, which may be due to the high filler percentage (59.29 wt.% in unaged samples calculated by TGA mass loss) in epoxy composites. The properties that changed the most with accelerated degradation tests were tensile strength and ester C=O absorption. According to the IEC 60505 standard, the tensile strength was used as the criterion of the lifetime prediction model for epoxy composites [45].

### 3.2. Tensile Strength

Figure 4 exhibits the relative tensile strengths of aged samples under different aging conditions. After curve fitting, three equations were obtained to describe the correlation between the degree of tensile strength degradation and aging time. The lifetime under each aging condition was estimated by calculating the x value when the tensile strength dropped to 50% of its initial value according to IEC 60216-2 [60]. The original relative tensile strength (aging time is zero hours) for 3 conditions (75 °C and 95% RH, 85 °C and 95% RH, 95 °C and 95% RH) was 100%. After curve fitting, the equations show the different values for each condition at zero-time aging. Predicted tensile strength values at time zero are inaccurate due to the lack of data during early aging. However, our study was designed to predict lifespan after long-term aging, so we focus more on the data at longer hours.

According to ISO 11346, for the temperature dependant degradation process, the lifetime/temperature relationship can be represented by the Arrhenius equation. The lifetime of the examined epoxy composites is given by ln *t* = −E/RT + A [61], in which *t* is the life expectancy, A is a constant, T is the thermodynamic temperature, E is the activation energy (J/mol), and R is the gas constant (8.314 J/mol/K). Based on the lifetime prediction model shown in Figure 5, −E/R = 8992.6 (K), the activation energy can be obtained as 74.76 kJ/mol or 0.77 eV and A = −17.582. Therefore, life expectancy at a specific temperature and 95% RH can be predicted by extrapolating data using log life versus 1000/T coordinate plotting. Table 3 lists the calculated lifetimes of epoxy composites at selected temperatures and 95% RH. The lifetime was sensitive to the aging temperature: the calculated lifetime was approximately 33.16 years at 25 °C and 95% RH, while it did not exceed half a year at 75 °C and 95% RH.

### 3.3. Ester C=O Absorption

As the accelerated aging time increased, the ester bonds of epoxy resin were destroyed by water molecules at high temperatures (Figure 2). Figure 6 shows the relative ester C=O absorption on the surface of aged samples at different temperatures as a function of aging time. The original C=O absorption is the same for different tests. After curve fitting, the equations estimate the different C=O absorption values for each condition at zero-time aging. Predicted C=O absorption values at time zero are inaccurate due to the lack of data during early aging. However, our study was designed to predict lifespan after long-term aging, so the authors did not focus on these data points. The curves decreased with aging time and were inversely related to temperature. Since both the relative ester C=O absorption and tensile strength displayed a distinctive declining trend with the aging test, a correlation can be established between these two indicators to predict the point of material failure. The results are shown in Table 4. For aging at 75 °C and 95% RH, samples failed at 3696.9 h and the average ester C=O absorption dropped to 32.80% of that of the unaged samples. For aging at 85 °C and 95% RH, samples failed at 1987.2 h and the average ester C=O absorption dropped to 28.14% of that of the unaged samples. For aging at 95 °C and 95% RH, samples failed at 907.0 h and the average ester C=O absorption dropped to 22.68% of that of the unaged samples. In summary, at the material failure point, the tensile strength of epoxy composites dropped to 50% and the ester C=O absorption of epoxy composites dropped to 27.87 ± 5.07%. This correlation can be used as an alternative method for assessing the “health” of insulation materials during on-site inspections, since FTIR is a non-destructive method and portable versions of this equipment are commercially available. As the C=O bond absorption of epoxy composites is strongly correlated with their tensile strength, it can provide a convenient and non-destructive method for on-site monitoring of the state of Huntsman epoxy composite system #229-based insulation materials. The mechanism of the degradation process will be discussed later to validate the proposed method.

The ester C=O absorption model is different from the tensile strength model. The reason for this may be because tensile strength is a bulk property, while C=O absorption is measured from surface only. When the aging time increases, the surface property decreases faster than the bulk property, which leads to a liner relationship between tensile strength and aging time and an exponential relationship between C=O absorption and aging time. All R^2^ values are in the range of 0.7582–0.9541, and possible reasons for this are experimental errors during sample synthesis, aging, and testing. To evaluate the accuracy of the two aging models, actual aged samples are required in order to obtain their mechanical strength and C=O absorption. We can then predict their aging time using the two aging models. Finally, we can evaluate the models by comparing the predicted aging time to the actual aging time. The epoxy composite insulator lifetime for transmission and distribution network equipment was around 30 years [62], which is near to our predicted lifetime of 33.16 years at 25 °C and 95% RH. But the actual aging conditions are complicated by climate and load, and the average state is not necessarily 25 °C and 95% RH. The reduction in mechanical strength is a criterion specified in the IEC standard. However, this kind of destructive test cannot be carried out to evaluate the aging condition of insulating materials of electrical equipment in use; that is, it is not allowed to destroy the equipment in operation to evaluate its remaining service life. Therefore, in this work, we propose a non-destructive method—ATR-FTIR—to monitor the C=O bond absorption feature.

### 3.4. Hydrophobicity

Figure 7 displays the static contact angles of rectangular samples on the epoxy-water-air contact line. The samples were aged at 95 °C and 95% RH for different times. As the aging time increased, the surface wettability changed from hydrophobic to hydrophilic. According to the subsequent FTIR analysis, this was due to the hydrolysis of ester bonds under high humidity and high temperature conditions (Figure 2), generating hydrophilic carboxylic acids and alcohols. The carboxylic acids further reacted with the calcium ions of fillers (CaSiO_3_) to form hydrophilic carboxylates.

### 3.5. FESEM Analysis

Figure 8 shows the FESEM photograph of dumbbell-shape specimen cross sections after tensile testing. Samples aged at 95 °C and 95% RH for 1000 h showed microcracks with a width of 50 nm–1 μm and a length of 20 μm–90 μm, as shown in Figure 8B. No microcracks were found on the unaged samples, as shown in Figure 8A. This is the microscopic manifestation of the weakening of the mechanical properties of epoxy composites after aging at high temperature and high humidity. Figure 8C displayed the polished cross section of tensile specimens aged at 95 °C and 95% RH for 1000 h, with all cracks passing through the resin-filler interface marked with red arrows. FTIR analysis confirmed hydrolytic decomposition of epoxy composites. As the result of hydrolysis of epoxy resin ester bonds, organic acids and alcohols were generated. The Ca^2+^ ions in the filler CaSiO_3_ could be separated under the action of organic acids. These changes can occur at the interface between epoxy resin and inorganic fillers and can be accelerated under multi-stress conditions (moisture, temperature, mechanical, and electrical stresses), leading to failure of the resin-filler interface [63]. In brief, resin-filler interface deterioration was the main part of insulation aging.

Figure 9 shows the FESEM image and element composition analysis of the cross section of the unaged sample. Unaged epoxy composites contained 59.29 wt.% filler based on TGA data, as shown in Appendix A. Through FESEM imaging (Figure 9A–C), the filler’s morphology was examined using the cured epoxy composites and purified fillers from the uncured epoxy composites. The fillers presented tabular crystals in lamellar and compact aggregates, which was consistent with previous studies of wollastonite by other researchers [64]. By analysing a set of FESEM images, the filler size range was found to be approximately 100 nm–100 μm. Their elemental composition was confirmed as CaSiO_3_ by EDS analysis (Figure 9D).

### 3.6. Mass Change

Figure 10 demonstrates the mass change in epoxy composites under different aging conditions. Logarithmic trendlines were obtained by curve fitting. Compared with the fitting curve of 85 °C and 95% RH, the R^2^ values of fitting curves at 75 °C and 95% RH and 95 °C and 95% RH are relatively low, indicating a weaker correlation. Possible reasons include limited data points and large variance in synthesized sample thickness (1.5–2.5 mm) due to thermal expansion of silicone rubber moulds during sample curing. However, all fitting curves showed a saturation trend after specific aging times. It is well known that the main cause of mass change is water absorption. Therefore, the water uptake of epoxy composites increases with aging time at early stages, and eventually reaches equilibrium.

### 3.7. Degradation Mechanism

Figure 11 shows the FTIR spectra of epoxy composites after aging for different times at 85 °C and 95% RH. The peaks at 3342, 1728, 1550, and 1508 cm^−1^ were, respectively, from the O-H bonds in carboxylic acids or alcohols, the ester C=O bonds, C=O bonds in carboxylates, and C=C aromatic rings [65,66,67]. The aromatic ring C=C bond at 1508 cm^−1^ did not change in all accelerated degradation tests, so we used it as a quantitative reference in this study. With increasing aging time, the ester bond absorption at 1728 cm^−1^ decreased, while the absorption of O-H bonds in acids or alcohols at 3342 cm^−1^ and the absorption of C=O bonds of carboxylates at 1550 cm^−1^ increased. Therefore, the aging process of epoxy composites started with the hydrolysis of ester bonds in epoxy resin under high humidity and high heat conditions, producing carboxylic acids and alcohols. The carboxylic acids further reacted with the calcium ions of fillers (CaSiO_3_) to form carboxylates, which damaged the resin-filler interface (Figure 8C) and increased the material hydrophilicity (Figure 7). Hydrolysis of epoxy resin ester bonds and formation of carboxylates disrupted the epoxy resin networks and resin-filler interface, weakening material mechanical strength and causing cracks to appear in cross sections of aged tensile specimens (Figure 8B). For electrical insulation materials in use, these cracks can open the path for heat, moisture, and electronic charges, leading to more serious failures [68,69].

## 4. Conclusions

In this work, a lifetime prediction model was developed that can be used to estimate the lifetimes of epoxy composites at different temperatures and high relative humidity (95%). FTIR analysis proved that accelerated aging destroyed the ester bonds of the epoxy resin networks. The ester C=O absorption of epoxy composites decreased to 28% when the tensile strength was reduced to 50% after accelerated degradation. The associated degradation mechanism was identified as the hydrolysis of ester bonds in epoxy resin at high humidity and high temperature, disrupting the epoxy resin networks and weakening their mechanical strength. Meanwhile, the Ca^2+^ in fillers reacted with the carboxylic acids produced by the degradation of the epoxy resin ester bond to form carboxylates, which destroyed the resin-filler interface and increased the material hydrophilicity. Finally, we propose the application of ATR-FTIR spectroscopy to more easily monitor the degradation of Huntsman epoxy composite system #229-based electrical insulation in situ.

## Figures and Tables

**Figure 1 polymers-15-02666-f001:**
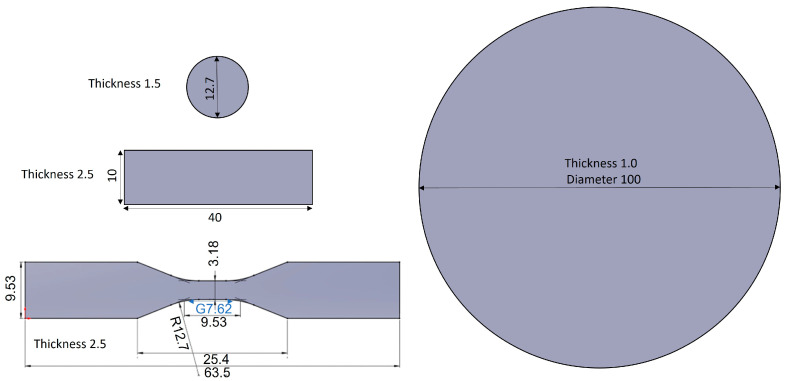
Schematic of the standard test samples including their dimensions (mm).

**Figure 2 polymers-15-02666-f002:**
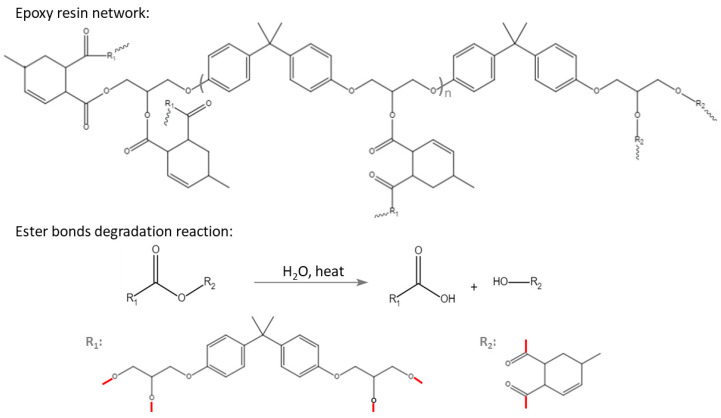
Chemical structure of DGEBA-anhydride epoxy resin networks and their ester bond degradation reaction under high temperature and high humidity. (The red lines are the bonding sites that connect to the epoxy resin network).

**Figure 3 polymers-15-02666-f003:**
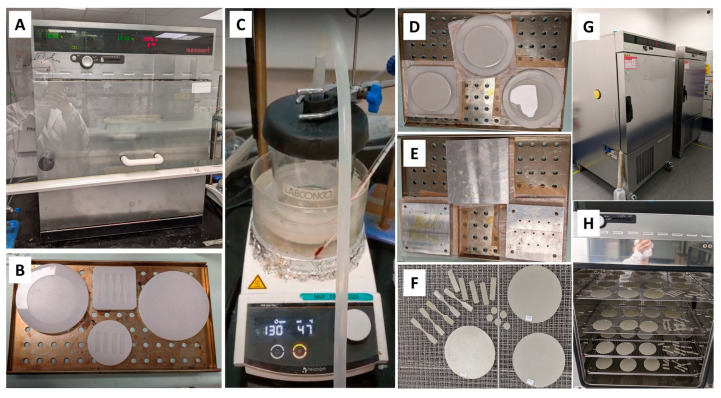
Epoxy composite sample preparation for the accelerated aging experiment: (**A**) Oven UNE 400 for curing, (**B**) Silicone rubber moulds, (**C**) Mix and degas the uncured epoxy composite system in a fast-freeze flask using a magnetic stirrer, (**D**) Cast the degassed mixture into preheated silicone moulds, (**E**) Cover the moulds with release films, weighing papers and metal blocks, (**F**) Demoulded specimens, (**G**) Climate chambers CTC256 for accelerated aging experiments, (**H**) Sample racks in the climate chamber.

**Figure 4 polymers-15-02666-f004:**
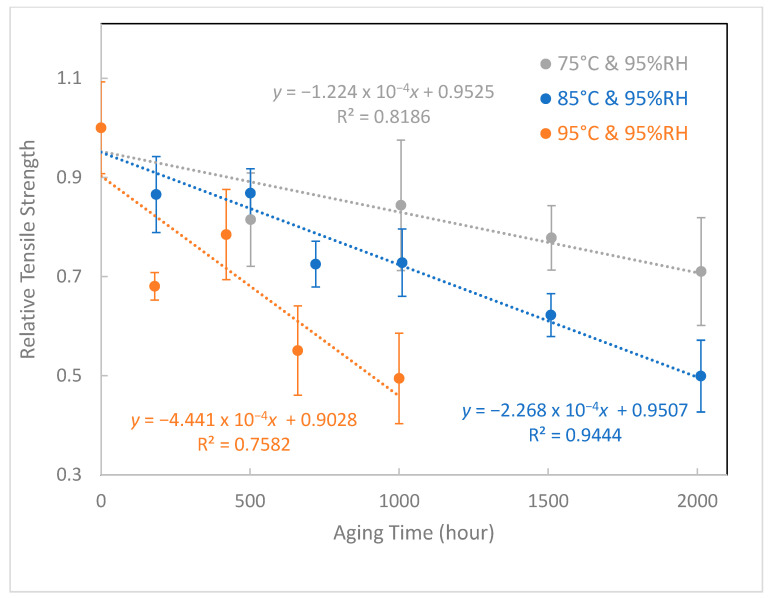
Relative tensile strength versus aging time at different aging conditions.

**Figure 5 polymers-15-02666-f005:**
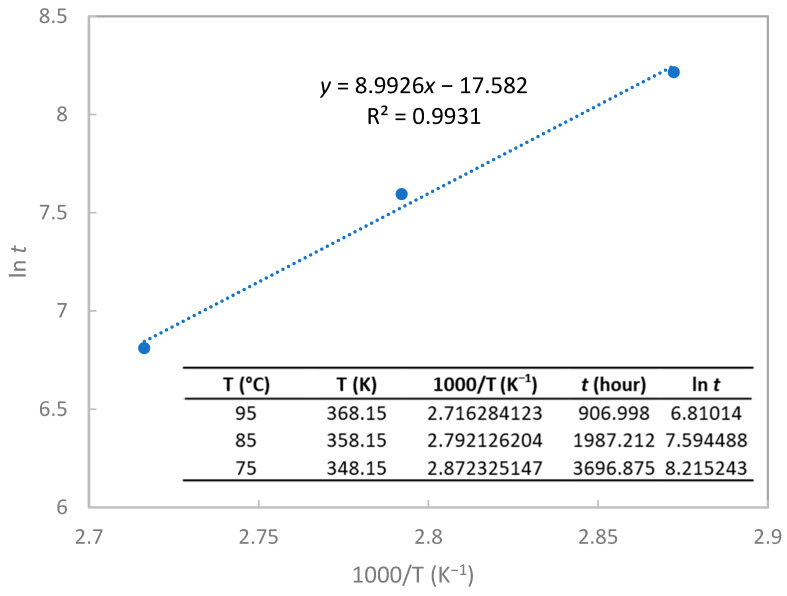
Lifetime prediction model for aging at different temperatures and 95% RH.

**Figure 6 polymers-15-02666-f006:**
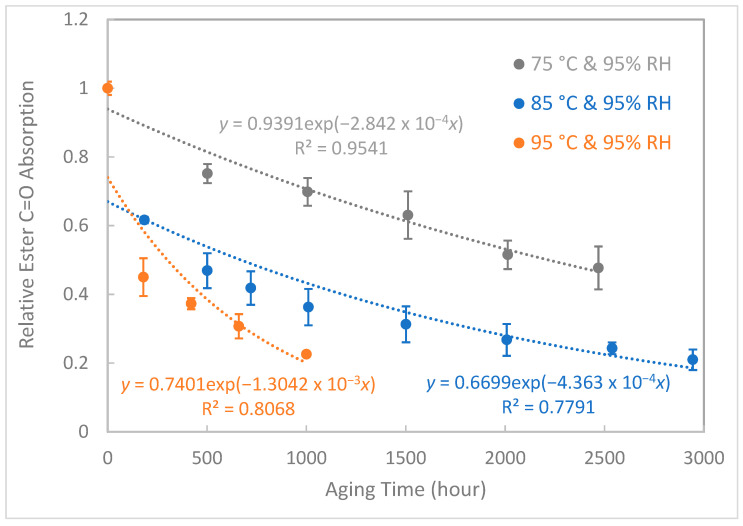
Relative ester C=O absorption versus aging time under different aging conditions.

**Figure 7 polymers-15-02666-f007:**
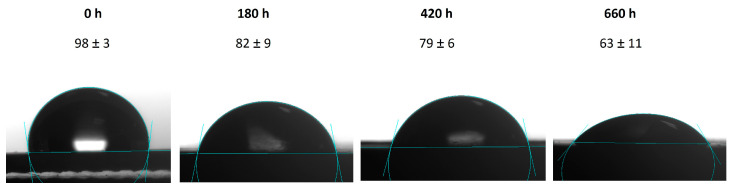
Contact angles of rectangular samples on the epoxy-water-air contact line.

**Figure 8 polymers-15-02666-f008:**
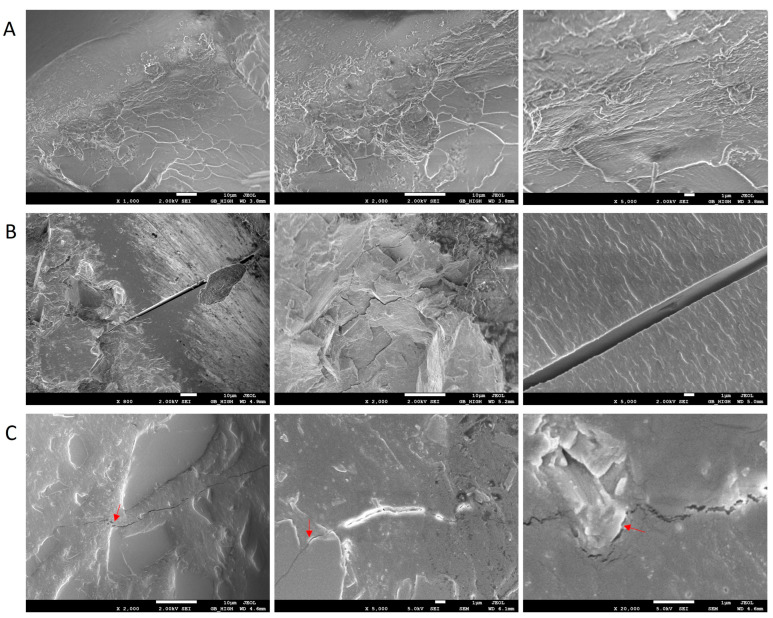
FESEM images of dumbbell-shaped specimen cross sections after tensile testing: (**A**) untouched cross section of unaged tensile specimens, (**B**) untouched cross section of tensile specimens aged for 1000 h at 95 °C and 95% RH, (**C**) polished cross section of tensile specimens aged for 1000 h at 95 °C and 95% RH (The resin-filler interfaces were marked with red arrows).

**Figure 9 polymers-15-02666-f009:**
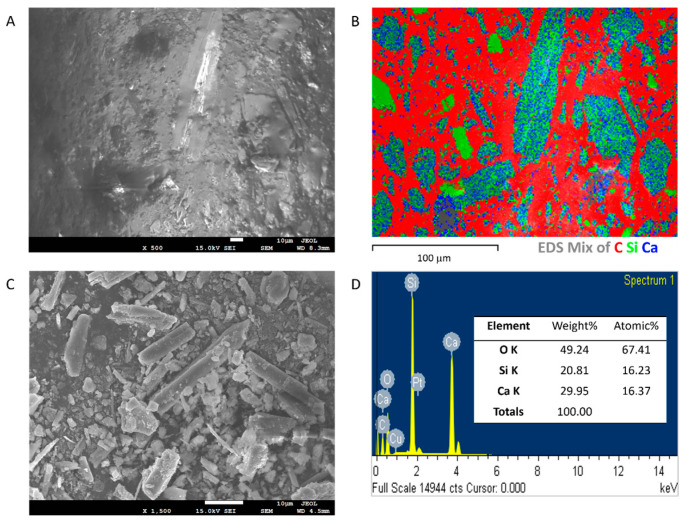
(**A**) FESEM image of a cross section of unaged epoxy composites. (**B**) EDS mapping of a cross section of unaged epoxy composites. (**C**) FESEM image of purified fillers from the uncured epoxy composites. (**D**) EDS analysis of purified fillers.

**Figure 10 polymers-15-02666-f010:**
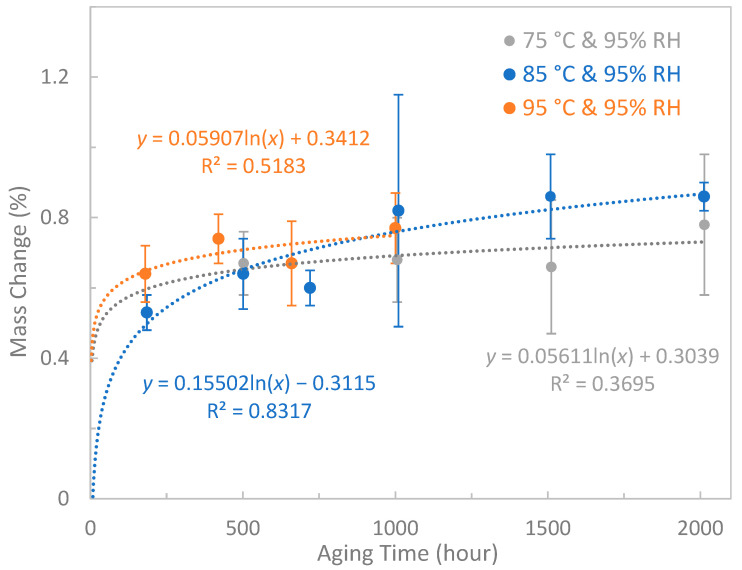
Mass change in epoxy composites under different aging conditions.

**Figure 11 polymers-15-02666-f011:**
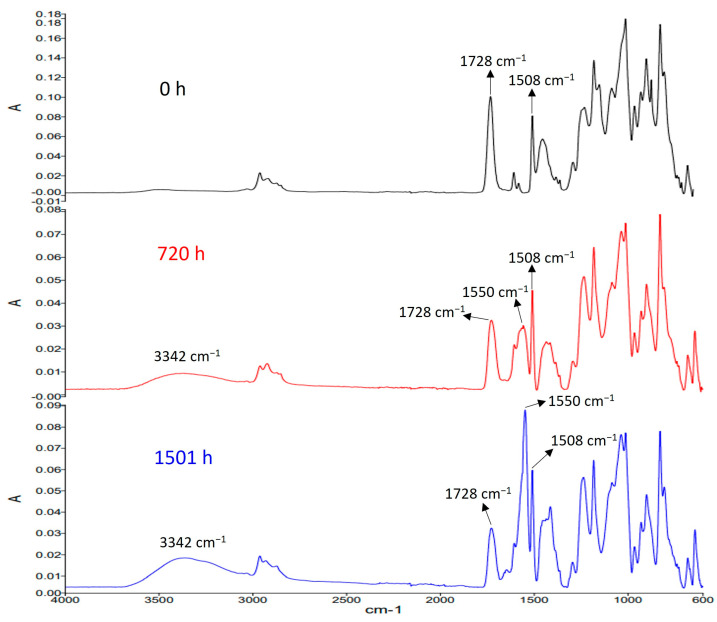
FTIR spectra of epoxy composites aged at 85 °C and 95% RH for different times.

**Table 1 polymers-15-02666-t001:** Material properties, characterization techniques, and related industrial standards for switchgear insulation material degradation studies.

Property	Industrial Standard	Characterization Technique
Tensile strength Elastic modulus	D638-14 [25] D229-19 [26]	Universal testing machine
Chemical structure	ISO 20368 [27]	Fourier transform infrared spectrometer (FTIR)
Thermal strain Glass transition temperature (Tg)	ASTM D7028-07 [28] ISO 6721 [29]	Dynamic mechanical analyzer (DMA)
Thermal expansion	ASTM E831-19 [30]	Thermomechanical analyzer (TMA)
Thermal conductivity	ISO 11357-8 [31] ISO 22007-1 [32] ISO 22007-4 [33]	Differential scanning calorimeter Modulated differential scanning calorimeter Laser flash
Heat capacity	ASTM E1269 [34] ISO 11357-4 [35] ASTM E1461-13 [36]	Differential scanning calorimeter Laser flash
Thermal stability	ASTM E2550 [37]	Thermogravimetric analyzer (TGA)
Contact angle	IEC 62073 [38]	Contact angle goniometer
Filler and microcrack	ASTM C1723 [39]	Field emission scanning electron microscope—energy dispersive spectrometer (FESEM-EDS)
Breakdown strength	IEC 60243-1 [40]	AC dielectric breakdown tester
Electric conductivity	IEC 62631-3-1 [41] IEC 62631-3-2 [42] IEC 62631-3-4 [43]	Electrometer/high-resistance meter
Dielectric loss	IEC 62631-2-1 [44]	Dielectric loss

**Table 2 polymers-15-02666-t002:** Accelerated aging conditions of epoxy composites.

Conditions	Aging Hours
95 °C and 95% RH	0, 180, 420, 660, 1000
85 °C and 95% RH	185, 501, 720, 1010, 1509, 2012
75 °C and 95% RH	502, 1006, 1511, 2013

**Table 3 polymers-15-02666-t003:** Lifetime prediction of epoxy composites at various temperatures and 95% RH.

Lifetime Prediction	*t* (year)
75 °C and 95% RH	0.44
65 °C and 95% RH	0.94
55 °C and 95% RH	2.10
45 °C and 95% RH	4.98
35 °C and 95% RH	12.46
25 °C and 95% RH	33.16

**Table 4 polymers-15-02666-t004:** Relative ester C=O absorption at failure points for different aging conditions.

Aging Condition	Lifetime (hour)	Relative Ester C=O Absorption	Change from Unaged
75 °C and 95% RH	3696.9	0.32795	−67.21%
85 °C and 95% RH	1987.2	0.28148	−71.85%
95 °C and 95% RH	907.0	0.22675	−77.33%

## Data Availability

Date are available from the corresponding author upon reasonable request.

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
