# Peer review of "Degradation and Lifetime Prediction of Epoxy Composite Insulation Materials under High Relative Humidity"

_polymers, 2023, doi:10.3390/polym15122666_

Round 1

Reviewer 1 Report

This manuscript investigated insulation failure of composite epoxy insulation materials in distribution switchgear under the stress of heat and humidity and a lifetime prediction model is developed. This work is important for researchers in electrical engineering field and it can be accepted for publication after major revision.

1. 2.1.6 and 2.1.7 are the same, which should be revised by the authors.

2. Fig.1,is it true that mechanical strength should be tensile strength?

3. Why original relative mechanical strength (ageing time is zero hours) for 95C and 95RH is 90%, while the other two cases are the same and higher?

4. Table 2, the results are for Huntsman epoxy resin #229, or for epoxy composites?

5. Fig.3, is it possible that the original C=O absorptions for different tests are different? The authors are expected to explain this results in Fig.3.

6. Discussion is actually Conclusion.

7. Discussion part, the lifetimes of Huntsman epoxy resin #229 is got. However, in the title, it is about “Lifetime Prediction of Composite Epoxy Insulation Materials”. Are epoxy resin and composite epoxy the same thing? It seems that the authors dont know the difference between epoxy resin and epoxy composites, which is also shown in Table 2.

8. Mechanical strength and C=O are used as indicators of aging prediction models, respectively, but there are some difference. Besides, Figure 1 shows a linear relationship between mechanical strength and aging time, while Figure 3 shows an exponential relationship between C=O and aging time, and all R2 are not high, indicating a week correlation. So how do the authors evaluate the accuracy of the two aging models?

9.The authors mentioned that there is a strong correlation between mechanical strength and C=O, how to reflect the relationship between them? The authors plan to use infrared detection to indicate aging state, so can mechanical strength be used as an auxiliary method to improve the accuracy of infrared prediction of aging life?

10.The author should give the formula referenced in the standard IEC 60216-2 and the specific procedure for calculating A and B in the Arrhenius formula.

Reviewer 2 Report

The below comments must be addressed n details before publication of the manuscript.

It seems that table 1 represents the general standards for insulation materials not the standards for testing and analysing the ageing process and degradation mechanisms in these materials.

The introduction needs to be revised by adding more relevant research recently published such as the below ones regarding the ageing process of epoxy components and degradation mechanisms in epoxy based materials.

Indentation characterization of glass/epoxy and carbon/epoxy composite samples aged in artificial salt water at elevated temperature

The Aging Behavior and Life Prediction of CFRP Rods under a Hygrothermal Environment

The novelty of the work is not clear. It must be highlighted in the last paragraph of the introduction.

Some images of preparation process, ageing process, schematic of the components, etc. Should be added to the manuscript to help the readers to understand the process.

schematic of the standard tests samples including their dimensions must be added.

Page 4, line 131: " For electrical insulation materials, the maximum test temperature should be below their transition temperatures including glass transition, melting, boiling and crystallization". Must it be less than Tg or less than Tg minus a specific value (for example 10oC)? 

Table S1 shall be included the main text of the manuscript.

The section "2.1.7. Contact Angle" has an irrelevant text and needs to be revised.

The curves regarding water uptake over time must be included in the main body of the paper.

Take a look at this recently published papers in terms of life prediction of aged composite materials, It is worth it to discuss this in your work and mention the other non linear life prediction models.

Move Figures S1, S2 and S3 to the main body of the manuscript. 

How did you measure the relative ester C=O absorption on the surface? Explain.

Discussion section is too poor. Discuss the results in details.

A conclusion section including the main achievements of the work is necessary.

Round 2

Reviewer 1 Report

All my concerns were addressed by the authors, and the manuscript was revised clearly in detail. I think this manuscript now can be accepted for publication.

Reviewer 2 Report

The paper is accepted in its current form.